# 🎨 HYBRID-GYM: Training Coding Agents to Generalize Across Tasks

**Yiqing Xie** [1]  **Emmy Liu** [1]  **Gaokai Zhang** [1]  **Nachiket Kotalwar** [1]  **Shubham Gandhi** [1]  **Sathwik Acharya** [1]
**Xingyao Wang** [2]  **Carolyn Rosé** [1]  **Graham Neubig** [1 2]  **Daniel Fried** [1]

## Abstract

When assessing the quality of coding agents, predominant benchmarks focus on solving single issues on GitHub, such as SWE-Bench. In contrast, in real use these agents solve more various and complex tasks that involve other skills such as exploring codebases, testing software, and designing architecture. In this paper, we first characterize some transferable skills that are shared across diverse tasks by decomposing trajectories into fine-grained components, and derive a set of principles for designing auxiliary training tasks to teach language models these skills. Guided by these principles, we propose a training environment, HYBRID-GYM, consisting of a set of scalable synthetic tasks, such as function localization and dependency search. Experiments show that agents trained on our synthetic tasks effectively generalize to diverse real-world tasks that are not present in training, improving a base model by 25.4% absolute gain on SWE-Bench Verified, 7.9% on SWT-Bench Verified, and 5.1% on Commit-0 Lite. HYBRID-GYM also complements datasets built for the downstream tasks (e.g., improving SWE-Play by 4.9% on SWT-Bench Verified)[1].

## 1. Introduction

Recent advances in large language models have enabled LM-based agents (Yang et al., 2024; Wang et al., 2025) to tackle real-world coding tasks, with rapid progress in benchmarks such as the SWE-Bench issue resolution benchmark (Jimenez et al., 2024). While most existing methods only focus on the issue-solving task (Pan et al., 2024; Jain et al., 2025b; Yang et al., 2025c), the application of coding

agents can be diverse in real-world usage. A recent study show that more than 70% of the real-world prompts target other software engineering tasks (Chen et al., 2025).

As shown in a concurrent work (Zhu et al., 2025), training on a single task can lead to overfitting and is not sufficient for generalizing to general coding agent tasks. In particular, agents trained on issue solving do not transfer reliably to tasks such as test generation or library implementation. This calls for training coding agents with stronger **task generalization** capabilities. Yet, in the current literature, task transferability for coding agents remains underexplored. In this work, we ask: what commonalities do real-world coding tasks share? How can we design training tasks that transfer effectively across a broad range of coding agent tasks?

To answer these questions, we start by decomposing real-world tasks into intermediate components (e.g., reasoning and repo-exploration) and analyzing the model's capabilities on each. These results are shown in Figure 1. We observe that a large number of actions in successful trajectories are spent on reasoning, repo-exploration, and implementation. These categories also remain challenging for current coding agents. For instance, even after finetuning on issue-solving (Pan et al., 2024), the agent still has file editing failures for the majority of examples.

Fortunately, these intermediate components can be isolated from the construction of coding environments with executable repositories (e.g., with all Python packages required by the repository installed). In previous work, setting up executable repositories is often viewed as a prerequisite for constructing training examples. This typically involves a series of complex steps such as package installation and test generation, which is still challenging for agents (Zhu et al., 2025) and time-consuming for humans (Pan et al., 2024; Yang et al., 2025c; Jain et al., 2025b). Based on the observation from our analysis, we ask: **can we design a set of tasks that cover the major components for coding practice: reasoning, repo-exploration, and implementation, but do not require executable repository setup?**

In this paper, we present HYBRID-GYM, a large-scale coding agent training dataset containing a suite of coding agent training tasks that only require simple environment setup. Based on our analysis summarized in Figure 1, we present

---

[1]Carnegie Mellon University [2]All Hands AI. Correspondence to: Daniel Fried <dfried@cs.cmu.edu>, Yiqing Xie <yiqingxi@cs.cmu.edu>.

*Proceedings of the 43rd International Conference on Machine Learning*, Seoul, South Korea. PMLR 306, 2026. Copyright 2026 by the author(s).

[1]Code: `github.com/Hybrid-Gym/Hybrid-Gym`

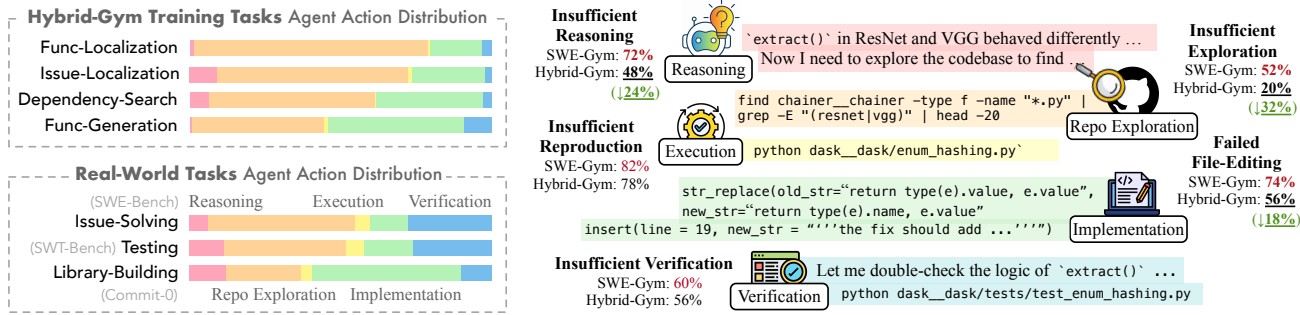

Figure 1. (*Left*) We decompose general coding agent tasks into a set of intermediate components and compute the percentage of agent actions spent on each component. Our training tasks partially cover verification and fully cover reasoning, repository exploration, and implementation, which consist of around 68% of the actions. (*Right*) Example actions for each component. Compared to a baseline training method, SWE-Gym (Pan et al., 2024), training with our HYBRID-GYM significantly reduces failures due to insufficient reasoning, insufficient exploration, and failed file editing, improving the final resolved rate on SWE-Bench Verified from 20.6% to 32.4%

a list of principles for training task design: the task needs to (1) match the output format of downstream tasks; (2) include a repository exploration process; and (3) require non-trivial reasoning. For scalability concerns, we further require that the task (4) must *not* require a complicated environment setup. Example tasks include function localization, issue localization, dependency search, and function generation. As shown in Figure 1, even without setting up executable repositories, all our tasks cover the major intermediate components of multiple downstream tasks, and such components are achieved with a similar set of actions across tasks. After training on HYBRID-GYM, the agent has significantly lower failure rates in reasoning, exploration, and file-editing compared to the baseline training method (Pan et al., 2024).

We conduct experiments on three real-world coding agent tasks: SWE-Bench issue-solving (Jimenez et al., 2024), SWT-Bench test generation (Mündler et al., 2024), and Commit-0 library generation (Zhao et al., 2025). Without training on any of the downstream tasks, HYBRID-GYM demonstrates strong task transferability on all three tasks, improving Qwen2.5Coder-32B by 25.4% / 7.9% / 5.1% on SWE-Bench Verified / SWT-Bench Verified / Commit-0 Lite, which achieves comparable performance to in-domain datasets (i.e., data built for the downstream tasks). Additionally, we also improve the performance of training on existing in-domain datasets by combining with HYBRID-GYM (e.g., we improve SWE-Play (Zhu et al., 2025) by 2.4% / 4.9% on SWE-Bench Verified / SWT-Bench Verified).

To provide insights to future research on coding agent training, we conduct a series of controlled experiments to reveal what transfers and what does not for coding agents. In addition to validating our task design principles that require output format matching, repo-exploration, and non-trivial reasoning, we observe that even successful trajectories of the same set of instances can lead to markedly different impact. Repository diversity improves training effectiveness, but training on the same repositories used in evaluation does

not inflate performance gains. The results emphasize the importance of teacher model selection and data selection.

## 2. HYBRID-GYM: Constructing Scalable Multitask Coding Agent Training Data

In this section, we first guide training task design by analyzing the commonalities shared among coding tasks and agent capabilities (§2.1). Based on the analysis results, we introduce a list of task design principles (§2.2) and present the four HYBRID-GYM tasks that satisfy all the principles (§2.3). Finally, we conduct a comparison between our training tasks and real-world coding tasks (§2.4).

### 2.1. Analysis of Coding Agent Task Transferability

With the ultimate goal of designing training tasks that transfer to diverse downstream tasks, we first investigate the commonalities shared by real-world tasks by answering: [**RQ1**] What are the commonalities of the components (i.e., high-level problem-solving steps) to complete real-world coding tasks? [**RQ2**] What are the commonalities of the concrete actions (i.e., commands) taken by the agents? Since effective training must target capabilities the agent has not yet learned, we also ask [**RQ3**] What are the abilities that the agent has not learned but are required for task completion?

**Analysis Setup**. Following prior works (Pan et al., 2024; Jain et al., 2025b; Zhu et al., 2025), we adopt the rejection sampling finetuning setting, where we finetune Qwen2.5-Coder-7B/32B on the successful trajectories generated for the training tasks. We conduct the analysis based on 200 successful trajectories sampled from each task, which are generated by Claude-Sonnet-4.5 (Anthropic, 2025b) using OpenHands as the agent scaffold (Wang et al., 2025)

**Task Analysis: Decomposition of Coding Tasks**. To answer **RQ1**, we first decompose general coding agent tasks into five intermediate components: reasoning, repo-

| Command | Example | Func-Localize | Issue-Localize | Dep-Search | Func-Gen | SWE-Bench | SWT-Bench | Commit-0 |
|---------|---------|---------------|----------------|------------|----------|-----------|-----------|----------|
| grep | grep -i "pipeline" | 31.3% | 36.1% | 48.4% | 89.9% | 35.4% | 37.4% | 5.3% |
| find | find . -type f -name "least_angle.py" | 17.5% | 11.1% | 21.4% | 8.7% | 16.2% | 21.5% | 31.6% |
| cd | cd /workspace/scikit-learn | 21.9% | 31.5% | 2.0% | 0.0% | 26.7% | 0.0% | 5.3% |
| ls | ls -la | 6.6% | 2.3% | 3.7% | 0.0% | 2.4% | 12.4% | 36.8% |

*Table 1.* The use of Linux commands for repository exploration in our proposed training tasks (left) and downstream tasks (right). We show the percentage of each command used in the agent steps targeting repository exploration.

exploration, execution of existing code, solution implementation, and verification. We then use o3-mini (OpenAI, 2025b) to categorize each agent action by the component it targets. We manually examined 20 cases and made the same categorization as o3-mini in all the cases.

Results in Table 6 show that (1) The action distribution across different components is similar for the three downstream tasks, with repo-exploration accounting for the largest share. (2) We notice that around 70% of the actions fall into categories that do not require setting up executable repositories: reasoning, repo-exploration, and implementation. This is notable because executable repository setup is complicated and challenging for both humans and agents (Yang et al., 2025c; Zhu et al., 2025) and is the bottleneck of constructing scalable training datasets. Our analysis hence suggests a key direction: training tasks that avoid executable repository setup can also be beneficial.

**Task Analysis: Concrete Actions Used in Coding Tasks**. To answer **RQ2**, we analyze the concrete agent tools used in each task. The tools built for the OpenHands agent (Wang et al., 2025) include bash command execution (`execute-bash`), file viewing (`view`), and file editing (`str-replace`). Results in Table 6 show that all three tools are heavily used across downstream tasks. This suggests that *the agents need to learn all the tools in training.*

For the bash execution tool, we further analyze the specific commands issued by the agent. Table 1 shows that different tasks rely on highly similar repo-exploration commands, such as grep, find, cd, and ls, suggesting that *repo-exploration skills could be broadly transferable across tasks.*

**Agent Capability Analysis**. To answer **RQ3**, under the setting of distillation-based training, we conduct an error analysis for the untrained student model (Qwen2.5Coder-7B, Hui et al. (2024)), the student model trained on a baseline dataset (SWE-Gym, Pan et al. (2024)), and the teacher model (Claude-Sonnet-4.5, Anthropic (2025b)). We define error categories along two axes: (1) failures in the intermediate components identified in our previous analysis, and (2) failures in the use of each tool. We then prompt an LLM to label whether each agent trajectory exhibits these failures.

Figure 5 presents the error analysis results. We observe several error categories where the fine-tuned model's failure rate remains far higher than the teacher model's: insufficient repo-exploration, insufficient reasoning, and failure/loop on

the file-editing tool. This suggests that *these capabilities still have substantial room for improvement.*

## 2.2. Principles for Training Task Design

We summarize the analysis results in §2.1 as follows: (1) reasoning, repo-exploration, and implementation are the dominant intermediate components shared across coding-agent tasks; (2) The concrete types of commands to complete these components are also similar across tasks; and (3) The performance gaps between the student and teacher models are still large for the above categories and file editing.

Based on the task commonalities and the agents' capabilities, we present the following principles for training task design:

[**Principle 1**] The tasks should have the same output format as the downstream tasks. For the downstream tasks in our evaluation, the format is generating a code patch in the codebase, which requires file editing. A counterexample for training tasks is a variant of our issue-localization task, where we only need the agent to generate the plan to resolve the issue in the message, without making any code changes.

[**Principle 2**] The tasks should involve a meaningful repository-exploration phase. A counterexample is a script-level code generation task (e.g., HumanEval (Chen et al., 2021)). Although we can still provide a repository as context by putting the solution template file inside an empty repository, it does not involve any file localization.

[**Principle 3**] The tasks should require a non-trivial level of reasoning to complete. Namely, the agent must make substantive decisions, such as choosing the problem-solving strategy, generating the precise tool call arguments, and writing the code. A counterexample is a simple string-replacement task where we provide the exact edit location and the old/new strings. The agent can succeed by a single `str-replace` call without inspecting the codebase.

With the observation that all the major components: reasoning, exploration, and implementation, do not require an executable repository, we further require the training tasks to be scalable (i.e., easy to generate new instances):

[**Principle 4**] The task setup should not require substantial effort. For instance, the task should avoid installing packages for the entire repository or generating executable test files, which are challenging for both humans and agents.

In §3, we will show that all HYBRID-GYM tasks satisfy all

four principles and generalize well to multiple downstream tasks. In contrast, §4 will show that all three counterexample tasks mentioned above do not yield effective task transfer.

## 2.3. HYBRID-GYM Tasks

In this section, we introduce four HYBRID-GYM tasks that satisfy all the principles above. We provide the prompt for each task in Appendix A.1.

**Function Localization**. To boost repository exploration (principle 2), we design a task to localize a function in the entire codebase: given a short description, the agent needs to identify the corresponding function in the codebase. To satisfy Principle 1, which requires generating a code patch as the outcome, we require the agent to write a docstring containing the fix plan. We evaluate the task by checking whether a docstring is added for the correct function. This task requires substantial reasoning (principle 3), as the agent needs to understand the code content and conduct an effective and targeted search over the entire repository, which contains 2,925 functions on average in our training set.

**Issue Localization**. Function localization aims to locate the code based on the description, and issue localization locates the problematic code based on an issue. Given a GitHub issue, the task is to locate the relevant code in the repository and leave a comment containing a plan to fix the issue. We evaluate the task by checking whether a comment is generated in the same file where the actual fix is located.

Compared to the issue-solving task used in most prior work, constructing issue localization data does not need test execution for evaluation and can be instantiated from arbitrary GitHub issues. However, it still requires reasoning effort on analyzing the issue, exploring the repository, planning the fix, and making successful file-editing actions.

**Dependency Search**. We have the third task that targets code localization, which requires file-editing, repo-exploration, and reasoning: given a function, the agent must identify all functions or classes it directly calls. Similar to the above two tasks, we require adding comments to all these dependent modules. To obtain the ground truth, we adopt `Jedi`, a static Python analysis tool, to resolve imported names to their defining modules. Solving this task requires the agent to locate the target function, extract the referenced names, resolve each name to the correct module, and successfully edit the relevant files. This is non-trivial because a codebase may contain multiple modules with the same name, and only one corresponds to the actual call.

**Function Generation**. Beyond the localization tasks that only aim to generate comments, we design the function generation task that requires actual code generation. Given a function in a codebase, we use an LLM to generate a

description of its functionality. Then we remove the function body and ask the agent to re-implement it. To evaluate the implementation with test cases while avoiding repository installation, we adapt the RepoST (Xie et al., 2025b) method to the coding agent setting, which extracts the target function and its dependencies to a separate script and generates tests.

Here we highlight that generating *executable test scripts* is substantially easier and more cost-efficient than setting up *executable repositories*. To execute the test script, we only need to install the packages needed by the extracted function and its dependencies (2.1 packages on average), while installing the entire repository requires 26.5 packages on average. Furthermore, compared to *generating a new test file* that requires understanding the repository structure and resolving relative imports, it is substantially easier to *produce tests for a separate script*, which only requires seq-to-seq generation, without any agentic frameworks, making the setup process cost-efficient.

We emphasize that HYBRID-GYM is not limited to the specific tasks introduced here: other task designs may also satisfy our principles. It is also possible to construct variants of our tasks, such as localizing multiple functions, or finding the empty function before generating the function body. We leave the exploration to future works.

## 2.4. Verification of HYBRID-GYM Tasks

In this section, we verify that our training tasks share commonalities with real-world tasks and remain challenging for the student models we use: Qwen2.5Coder 7B and 32B.

**HYBRID-GYM tasks cover the majority of intermediate components in multiple real-world tasks**. As shown in Table 6, all the training tasks cover reasoning, repo-exploration, and implementation, and the function-generation task additionally includes solution verification. We note that although our training tasks do not explicitly include executing existing code, it typically accounts for only a small fraction of an agent's trajectory.

**HYBRID-GYM trajectories contain similar actions with real-world tasks**. As shown in Table 1, the training tasks use a similar set of bash commands for repo-exploration as the real-world coding tasks. Similarly, Table 7 shows that the training tasks cover the full set of OpenHands tools used in downstream tasks. Together, these results suggest that both the high-level problem-solving components and the concrete actions that achieve them can transfer across HYBRID-GYM tasks and downstream tasks.

**HYBRID-GYM tasks are still challenging for models without finetuning**. Under the setting of distillation learning, training tasks are more useful when there is a larger performance gap between the student and teacher models. As a

| Model | Func-Localize | | Issue-Localize | | Dep-Search | | Func-Gen | |
|---|---|---|---|---|---|---|---|---|
| | Success % | Non-Empty % | Success % | Non-Empty % | Success % | Non-Empty % | Success % | Non-Empty % |
| Qwen2.5Coder-7B | 0.0% | 50.0% | 3.3% | 16.7% | 6.7% | 60.0% | 5.0% | 70.0% |
| Qwen2.5Coder-32B | 40.0% | 95.0% | 33.3% | 83.3% | 56.7% | 86.7% | 25.0% | 95.0% |
| Claude-Sonnet-4.5 | 70.0% | 100.0% | 63.3% | 100.0% | 80.0% | 100.0% | 50.0% | 100.0% |

*Table 2.* Performance of the student and teacher models on a subset of instances sampled from each training task.

| Dataset | # Trajs | # Repos | Avg Steps | # Images | Cost |
|---|---|---|---|---|---|
| SWE-Gym | 491 | 11 | 40.2 | 2.4k | unknown |
| R2E-Gym | 3,321 | 10 | 33.2 | 4.5k | unknown |
| SWE-Smith | 5,016 | 128 | 26.7 | 128 | 2.32¢ |
| SWE-Play | 704 | 28 | 72.9 | 28 | unknown |
| HYBRID-GYM | 4,470 | 762 | 39.1 | 2 | 0.07¢ |
| Func-Localize | 1,438 | 226 | 29.0 | 1 | 0.02¢ |
| Issue-Localize | 1,978 | 263 | 52.4 | 1 | 0.00¢ |
| Dep-Search | 502 | 120 | 26.9 | 1 | 0.00¢ |
| Func-Gen | 552 | 306 | 28.6 | 1 | 0.56¢ |

*Table 3.* Statistics of HYBRID-GYM and its subsets. Following SWE-Smith, we report the average per-instance cost of setting up the training environment for rollout. Compared to existing datasets, HYBRID-GYM covers more repositories and requires only 2 docker images to build all training instances.

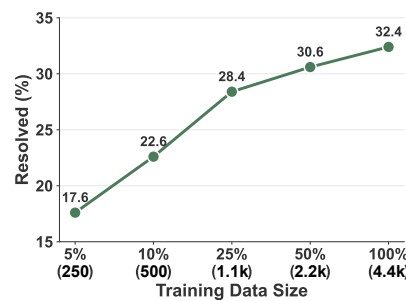

*Figure 2.* Scaling law analysis. Performance on SWE-bench Verified improves consistently as training data size increases from around 5% (250 trajectories) to 100% (4.4k trajectories).

sanity check, before generating training data, we sample 20-30 instances from each task and evaluate the teacher model (Claude-Sonnet-4.5) and student models (Qwen2.5Coder, 7B and 32B).

Table 2 show that Claude-Sonnet-4.5 achieves substantially higher success rates than Qwen2.5Coder-32B on all four tasks, with gaps ranging from 23.3% to 30.0%. This confirms that all tasks provide sufficient room for distillation.

## 3. Experimental Results

### 3.1. Experimental Setup

**Statistics of HYBRID-GYM**. Table 3 presents the statistics of HYBRID-GYM and existing datasets. Based on the scalable training tasks, we collect 4.4k trajectories from 762 repositories using only two Docker images. The environment construction cost is only 0.07¢per example, which is 16× lower than SWE-Smith (Yang et al., 2025c). Trajectory lengths vary by task, ranging from 26.9 to 52.4 agent steps, which are comparable to the issue-solving trajectories in prior datasets, suggesting that our tasks retain a meaningful level of complexity. The trajectories are obtained from the combination of Claude-Sonnet-4.5 (Anthropic, 2025b), Claude-Sonnet-3.7 (Anthropic, 2025a), and Qwen3-235b (Yang et al., 2025a). We provide dataset details in Appendix A.1.

**Evaluation Datasets and Metrics**. We follow prior work (Zhu et al., 2025) and report the resolved rates on SWE-Bench Verified (Jimenez et al., 2024), SWT-Bench Lite/Verified (Mündler et al., 2024), and Commit-0 (Zhao et al., 2025). On SWE-Bench, we additionally report the

localized rate: the percentage of code patches applied to the correct file, and the non-loop rate: the percentage of trajectories without any "loops", which is defined as repeating the same action three times consecutively (Pan et al., 2024).

### 3.2. Main Results

**HYBRID-GYM Shows Strong Task Transfer Performance**. Results in Table 4 show that HYBRID-GYM delivers strong task generalization performance across all three benchmarks. Relative to the base 32B model, it improves performance by 25.40 % / 7.85 % / 5.11 % on the three tasks, outperforming all existing task transfer training datasets. Notably, without training on the downstream tasks, HYBRID-GYM matches or even exceeds in-domain training. For example, SWE-Play-32B reaches 31.2 % / 16.17 % on SWE-Bench Verified / SWT-Bench Verified, whereas HYBRID-GYM-32B achieves 32.4 % / 16.86 %. This suggest that our tasks capture the commonality shared among general coding agent tasks and transfer effectively across all three downstream tasks. These results are especially compelling because HYBRID-GYM requires substantially less effort to build new training instances than prior datasets (e.g., as shown in Table 3, SWE-Smith requires 128 docker images and costs 2.32¢per example, while HYBRID-GYM is only built with 2 images and an average cost of 0.07¢).

**HYBRID-GYM Improves In-Domain Training Datasets**. Table 4 further shows that we can further improve in-domain datasets by combining with HYBRID-GYM. For instance, we improve SWE-Play by 2.40% / 4.85% / 1.57% on SWE-Bench Verified / SWT-Bench Verified / Commit-0 Lite. The gains from mixing HYBRID-GYM with in-domain data sug-

| Training Tasks | Training Dataset | Issue Solving (swe) SWE-bench Verified | | | Test Generation (swt) SWT-Bench Lite & Verified | | Library Build (cmt) Commit-0 Lite |
|---|---|---|---|---|---|---|---|
| swe swt cmt syn | | Resolved % | Localized % | Non-Loop % | Lite Resolved | Verified Resolved | Resolved % |
| | | *Base Model: Qwen2.5Coder-7B* | | | | | |
| | (*base*) Qwen2.5Coder-7B | 1.8 | 4.4 | 60.4 | 0.72 | 0.23 | 4.80 |
| ✓ | SWE-Play-general | 8.6 +6.8 | 49.2 +44.8 | 93.4 +33.0 | 2.54 +1.82 | 1.62 +1.39 | 9.25 +4.45 |
| ✓ | SWE-Gym | 10.6 +8.8 | 48.2 +43.8 | 79.0 +18.6 | 1.45 +0.73 | 0.69 +0.46 | 8.61 +3.81 |
| ✓ | R2E-Gym | **19.0 +17.2** | — | — | 0.72 +0.00 | 0.46 +0.23 | 9.25 +4.45 |
| ✓ | SWE-smith | 15.2 +13.4 | — | — | 0.00 -0.72 | 0.00 -0.23 | 9.32 +4.52 |
| ✓ ✓ ✓ ✓ | SWE-Play | 17.0 +15.2 | 60.4 +56.0 | 91.2 +30.8 | 3.26 +2.54 | 3.00 +2.77 | 10.42 +5.62 |
| ✓ | **HYBRID-GYM** | 15.0 +13.2 | **62.4 +58.0** | **97.6 +37.2** | 2.90 +2.18 | 2.54 +2.31 | 10.54 +5.74 |
| ✓ ✓ ✓ ✓ | **HYBRID-GYM** + SWE-Play | 17.6 +15.8 | 58.6 +54.2 | 97.2 +36.8 | **5.07 +4.35** | **4.39 +4.16** | **11.02 +6.22** |
| | | *Base Model: Qwen2.5Coder-32B* | | | | | |
| | (*base*) Qwen2.5Coder-32B | 7.0 | 45.6 | 70.6 | 9.42 | 9.01 | 8.34 |
| ✓ | SWE-Gym | 20.6 +13.6 | 57.8 +12.2 | 76.2 +5.6 | 3.26 -6.16 | 2.77 -6.24 | 9.58 +1.24 |
| ✓ | R2E-Gym | 34.4 +27.4 | — | — | 3.26 -6.16 | 4.39 -4.62 | 11.56 +3.22 |
| ✓ | SWE-smith | **40.2 +33.2** | — | — | 13.77 +4.35 | 10.62 +1.61 | 12.38 +4.04 |
| ✓ ✓ ✓ ✓ | SWE-Play | 31.2 +24.2 | 73.4 +27.8 | 85.4 +14.6 | 18.12 +8.70 | 16.17 +7.16 | 13.95 +5.61 |
| ✓ | **HYBRID-GYM** | 32.4 +25.4 | **75.4 +29.8** | 98.2 +27.6 | 17.03 +7.61 | 16.86 +7.85 | 13.45 +5.11 |
| ✓ ✓ ✓ ✓ | **HYBRID-GYM** + SWE-Play | 33.6 +26.6 | 75.2 +29.6 | **99.2 +28.6** | **20.29 +10.87** | **21.02 +12.01** | **15.52 +7.18** |

*Table 4.* Results on three real-world coding tasks. We show whether the datasets contain swe, swt, cmt, or synthetic tasks (syn) in training. Results with a white background are **task transfer** results. Results with a light grey background are trained with in-domain tasks. We highlight the best result for each dataset and show the relative gain and loss compared to the base model, Qwen2.5Coder. HYBRID-GYM achieves strong task transfer performances across all three benchmarks and further improves in-domain datasets (e.g., SWE-Play).

| Training Tasks (# Instances) | SWE-Bench Resolved % | SWT-Bench Resolved % | Commit-0 Resolved % |
|---|---|---|---|
| Qwen2.5Coder-7B | 1.8 | 0.23 | 4.80 |
| + Issue-Solving (491) | 10.6 +8.8 | 0.69 +0.46 | 8.61 +3.81 |
| + Func-Localize (500) | 12.8 +11.0 | **3.00 +2.77** | 8.74 +3.94 |
| + Issue-Localize (500) | 9.6 +7.8 | 2.31 +2.08 | 9.09 +4.29 |
| + Dep-Search (500) | 6.4 +4.6 | 2.31 +2.08 | 9.25 +4.45 |
| + Func-Gen (500) | 7.8 +6.0 | 2.54 +2.31 | 9.25 +4.45 |
| + **HYBRID-GYM**-full | **15.0 +13.2** | 2.54 +2.31 | **10.54 +5.74** |

*Table 5.* Performance of training Qwen2.5Coder-7B on each of the HYBRID-GYM tasks and the issue-solving task. We sample 500 instances from each of our tasks. Even without finetuning on issue-solving, function localization achieves a larger performance gain than SWE-Gym, which has 491 issue-solving instances. We report the resolved rates on SWE-Bench Verified, SWT-Bench Verified, and Commit-0 Lite.

gest that the two sources are complementary. HYBRID-GYM primarily teaches general agent skills (e.g., reasoning, repo-exploration, and reliable tool use), while in-domain data contributes task-specific behaviors (e.g., generating tests and building libraries from scratch). Combining them both leads to additive improvements.

### 3.3. Detailed Results

**Training on Individual Tasks**. To assess the contribution of each task, we sample 500 instances per task and train the model on each subset separately. This training size is comparable to the SWE-Gym issue-solving dataset, which contains 491 examples. As shown in Table 5, each task yields a substantial improvement over the base model, with gains ranging from 4.6% to 11.0%. The tasks emphasize different capabilities (e.g., function localization performs

best on SWE-Bench but worst on Commit-0), while the full HYBRID-GYM dataset—combining all four tasks—achieves the strongest overall performance.

Notably, at similar training scales, the function-localization and issue-localization tasks provide gains that are comparable to, or even exceed, in-domain training. For example, SWE-Gym achieves a 10.6% resolved rate and a 48.2% localized rate, whereas Func-Localize (500) reaches 12.8% and 53.6%. A possible explanation is that localization trajectories devote more steps to repo-exploration than issue-solving, which is one of the agent's primary bottlenecks.

**Scaling Law Analysis**. As shown in Figure 2, we train the 32B model on progressively larger subsets of HYBRID-GYM, where we proportionally sample training instances from each task. The performance continues to improve as the dataset grows and still shows improvements when we have 4.4k training trajectories. The results suggest the advantage of constructing highly scalable training datasets.

## 4. Analysis: on the Task Transferability of Coding Agent Training

We have established in §3 that the particular tasks included in HYBRID-GYM are effective for task generalization across diverse real-world coding benchmarks. In this section, we investigate: **what properties of these trajectories enable task transfer?** To answer this, we run controlled ablations that systematically modify trajectory structure and measure the resulting changes in performance.

As the ablations require repeated evaluation across many

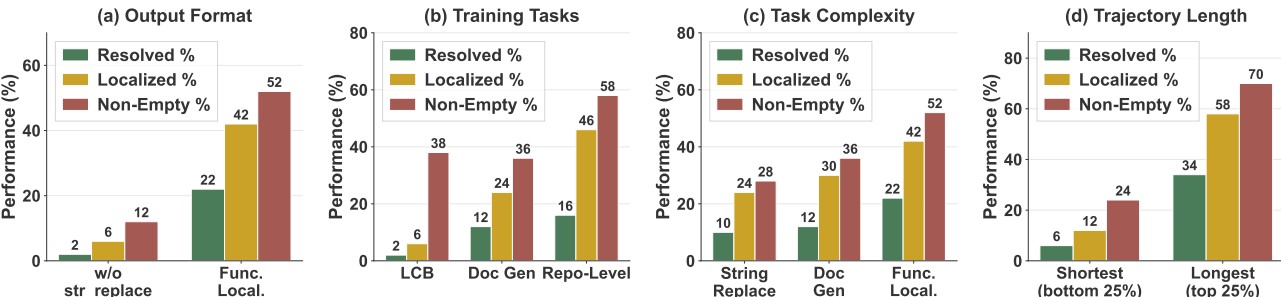

*Figure 3.* Controlled experiments on training data characteristics. (a) **Output Format:** Removing the file editing actions (`str_replace`) from function localization trajectories causes a large drop in SWE-bench resolution rate. (b) **Repo-Exploration:** script-level code generation (LCB) does not effectively transfer to repo-level issue-solving and even underperforms documentation generation, a simple repo-level task. (c) **Task Complexity:** Transfer improves as training tasks become more complex. (d) **Trajectory Complexity:** With a fixed data size, training on longer (more agent steps) trajectories substantially improves downstream performance.

variations, we keep the analyses computationally tractable by evaluating on an easier subset of SWE-bench examples (denoted as "Easy 50"), which are solvable by Deepseek-v3, a strong reference model (DeepSeek-AI et al., 2024).

## 4.1. Output Format Must Match Downstream Tasks

While our tasks are designed around higher level tasks such as function localization and generation, we also find that *how* these behaviors are expressed in the training trajectories matters. Specifically, as stated in [**Principle 1**] in §2, we find that the output format of the tasks is a crucial factor.

All three downstream tasks in our experiments require the agent to generate a code patch as the output. To isolate the role of output format, we conduct a minimal ablation on our function localization task: instead of requiring the model to generate a comment in the codebase containing the fix plan, we simply ask it to generate the plan in a message. We accordingly modify the training trajectories by removing the `str_replace` actions for writing comments and adding the fix plan in the final message generated by the agent.

As shown in Figure 3, performance on SWE-bench collapses when we remove the string replacement actions. These results suggest that for coding agent tasks, the production of **patch-like outputs** is not a superficial formatting choice but a crucial skill that the agent needs to learn.

## 4.2. Script-level Agentic Tasks Do NOT Generalize to Repo-level Tasks

We then validate [**Principle 2**], which requires the training tasks to have a repo-exploration stage. To test this, we implement a script-level task, LiveCodeBench (LCB) (Jain et al., 2024a), which primarily evaluates function generation in a single file. We contrast it with two repo-level tasks: (1) the repo-level function generation task we use in HYBRID-GYM (introduced in §2), and (2) documentation generation, which requires the agent to generate the docstring for a given function. For fair comparison, we wrap the coding

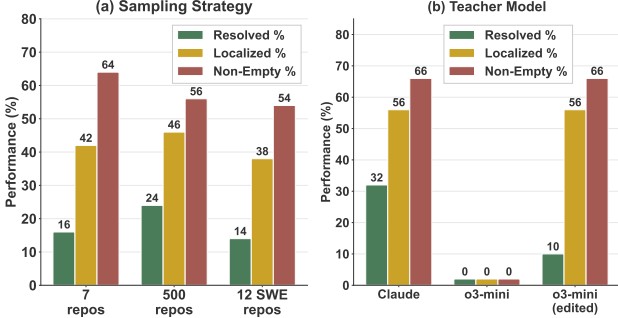

*Figure 4.* Effect of teacher model and data selection. (a) Effect of **sampling strategy** at fixed training budget. Repository diversity improves training but using the same repositories as in evaluation does not. (b) training on issue localization trajectories with different **teacher models**. o3-mini (edited) indicates the same set of trajectories as o3-mini, but with text and action steps combined.

template and test script in LCB in a dummy repository so that the model interacts with the same harness and tool API.

Figure 3 shows that despite having the wrapper to make LCB superficially a repo-level task, LCB is not effective for transfer to SWE-bench. This negative result suggests that repository-level generalization requires more than superficial "agentic formatting." In particular, repo-level tasks induce behaviors that are largely absent from script-level problems: identifying and navigating relevant files, grounding edits in existing codebase structure, and producing concrete patch-like modifications that apply cleanly in context.

## 4.3. Task and Trajectory Complexity Matters

After identifying reasoning and planning as a crucial component in coding tasks in §2, we derive [**Principle 3**] that requires the task to involve a non-trivial level of reasoning. Here, we examine whether task generalizability improves as training supervision more closely matches the multi-step, tool-mediated interaction patterns required for coding tasks.

We first examine tasks with ascending complexity: (1) single-shot string-replacement edits in which the task is

to refactor a function by simply renaming it without replacing other usages of it, (2) documentation generation for a fixed function (as in §4.2), and (3) the function localization task, which requires more complex reasoning than the previous two tasks. We see in Figure 3 that as tasks become more complex and interaction-heavy, downstream SWE-bench performance improves, and models are substantially more likely to produce non-empty patches.

To isolate complexity effects from task identity, we also investigate the effect of *trajectory complexity*, which is operationalized as the number of agent actions (tool calls and intermediate components) taken in successful trajectories. We pool trajectories from our repo-level task mixture and partition them by length, selecting the shortest and longest quartiles (25% each). As shown in Figure 3, although the training sizes are the same, training on longer trajectories yields a large improvement in SWE-bench resolution rate, alongside a sharp reduction in empty generations. These results suggest that trajectory complexity is an important driver of task performance.

### 4.4. Even Successful Trajectories of the Same Task Instances may have Different Impact

In addition to training task design, we observe that **the teacher model also plays a crucial role in effective training**. Specifically, we find that even successful trajectories for the same instances can yield very different distillation outcomes. We illustrate a particular case in Figure 4. On the issue localization task, we found that distilling from Qwen3-235B and Claude trajectories transfers well to SWE-Bench, but distilling from o3-mini trajectories degrades student performance. The key difference we found is that o3-mini frequently separates reasoning and actions (i.e., tool calls) into different turns. After training on such data, the agent collapses to the distribution of thinking-only conversational turns and does not learn to call tools at all. We edit o3-mini trajectories by simply stitching adjacent rationale and action steps into a single example. The resulting data effectively transfers to SWE-Bench, indicating that model-specific behaviors such as separating reasoning and actions can strongly influence the effectiveness of distillation.

### 4.5. Sampling Strategies Affects Effectiveness

Finally, beyond task choice and trajectory structure, we also find that how training instances are sampled impacts downstream transfer. We examine two key factors: (1) repository diversity (i.e., the number of repositories covered by the training set), which is identified by previous works (Xie et al., 2025b; Yang et al., 2025c), and (2) the domain of code. In the extreme case, we build training data from the exact same repositories we use in evaluation.

To investigate this, we keep the training budget fixed at 500 trajectories and sample from function localization trajectories. We compare: (i) *Min Repo-Diversity*, where we aggregate instances from a small number of large repositories (7 repositories), (ii) *Max Repo-Diversity*, where we sample one instance per repository (500 repositories) to maximize repository coverage, and (iii) *In-Domain Repos*, where we construct function localization trajectories based on the 12 SWE-Bench repositories.

Despite using the same number of training instances and the same SFT recipe, *Max Repo-Diversity* yields higher SWE-bench resolution (Figure 4). This suggests that exposure to a broader range of repository structures improves generalization to unseen repositories. Surprisingly, **training on the same repositories in evaluation does not improve the performance**, suggesting that learning general agent skills is more crucial than memorizing repo-specific information.

## 5. Related Work

**Coding Agents**. Compared to traditional seq-to-seq code generation (Chen et al., 2021) or static pipelines (Xia et al., 2024; Antoniades et al., 2025), coding agents can dynamically call tools to interact with the environment, enabling end-to-end problem solving. Recent works develop various coding agent scaffolds, including SWE-Agent (Yang et al., 2024), Live-SWE-agent (Xia et al., 2025) and Open-Hands (Wang et al., 2025) Such agentic methods achieve strong performance on diverse coding benchmarks, including issue-solving (Jimenez et al., 2024) under various settings (Jimenez et al., 2024; Deng et al., 2025; Yang et al., 2025b; Zan et al., 2025), test generation (Mündler et al., 2024; Jain et al., 2025a), and feature construction (Zhao et al., 2025; Zhang et al., 2023; Jain et al., 2024b; Xie et al., 2024; Ouyang et al., 2025). We therefore focus on improving agent systems for coding tasks.

**Coding Agent Pre-training**. In the pre-training stage, recent work train the language models on general agentic tasks, including proprietary models (OpenAI, 2025a; Anthropic, 2025b) and open-sourced ones (Yang et al., 2025a; DeepSeek-AI et al., 2024). For instance, Kimi-K2 (Team et al., 2025) constructs a wide range of synthetic tools and tasks to boost general agentic abilities. In this work, we focus on synthetic tasks for coding agents.

**Coding Agent Post-training**. In the post-training stage, one line of work focuses on the training method, such as rejection sampling (Pan et al., 2024) and reinforcement learning (Wei et al., 2025; Wang et al., 2024; Liu et al., 2026). Another line of work improves training data construction, yet most prior datasets only cover a single task such as issue-solving (Pan et al., 2024; Xie et al., 2025a; Ma et al., 2024; Jain et al., 2025b; Yang et al., 2025c), neglecting other crucial applications, including test generation and library con-

struction. A concurrent work, SWE-Play (Zhu et al., 2025), designs training data for each of the downstream tasks and additionally presents agent-proposed tasks. However, they do not conduct in-depth analysis on task transferability and fully rely on the agent to set up the complicated training environments, including building and installing a repository from scratch, which is challenging and limits the scalability. In contrast, we conduct systematic analyses on what transfers and what does not for coding tasks and design scalable and cost-efficient tasks based on analysis results.

## 6. Conclusion and Future Works

We conduct in-depth analyses of the transferability across coding agent tasks. We find that there are similar components shared by diverse real-world coding tasks (e.g., repo-exploration) that can be achieved by the agent with similar commands. Based on the discovery, we derive a set of principles for task design, including output format matching, requiring repo-exploration, non-trivial reasoning, and simple setup. We present four scalable and cost-efficient tasks that do not require setting up executable repositories. With the resulting dataset, HYBRID-GYM, we achieve strong task generalization results on three real-world benchmarks. For instance, even without issue-solving or test-generation data in training, we improve Qwen2.5Coder-32B by 25.4% on SWE-Bench Verified and 7.9% on SWT-Bench Verified. We also improve existing in-domain datasets by combining them with HYBRID-GYM (e.g., improving SWE-Play by 4.9% on SWT-Bench Verified).

We propose the following future directions:

**Exploration of training tasks that involve code execution**.. As mentioned in §2, other scalable training tasks may also satisfy our design principles. For example, a code execution task, verified by whether a function is successfully called with no execution error, or an environment setup task, which is evaluated by comparing the installed package versions with the setup file. Another possible task is to improve the function generation task by wrapping the evaluation script in an API call format and providing it to the agent during rollout. In this way, the agent can run the test to verify and improve its own solution. One can also make the current HYBRID-GYM tasks more challenging (e.g., also ask the agent to locate the function before implementing the function body).

**Analysis of trajectory effectiveness**.. We present in §4 that even successful trajectories of the same task instances may have substantially different impact in training. We use Figure 4 as an example. We encourage future researchers to explore the exact characteristics of a trajectory that are beneficial or harmful for training, and to improve the effectiveness of the trajectories for cost-efficient training.

## Acknowledgement

We thank Andre He, Abhav Mehrotra, Zora Zhiruo Wang, Pranjal Aggarwal, Jing Yu Koh, Saujas Vaduguru, Atharva Naik, Yuning Mao, Xuhui Zhou, and Tim Dettmers for their helpful feedback on this work. This work was supported in part by NSF grant DSES 2222762. EL was supported by the National Sciences and Engineering Research Council of Canada (NSERC), [funding reference number 578085], as well as the SoftBank-ARM Fellowship.

## Impact Statement

This paper presents work whose goal is to advance the field of Machine Learning, with an emphasis on improving the generalization of coding agents across diverse software engineering tasks, supporting more transferable automated programming systems and improved software maintenance. The potential societal impacts of this work are consistent with those of prior research on machine learning–based coding agents. We do not identify any ethical concerns or broader societal consequences that would otherwise require specific discussion.

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

# A. Appendix

## A.1. Dataset Construction Details

### A.1.1. FUNCTION LOCALIZATION

**Task**. Given a natural language description (without quick identifiers like names or paths) of a function , the agent must search the codebase to locate the target and add a docstring for it.

**Task Implementation**. Similar to issue localization, we build the tasks based on the SWE-Gym-Raw dataset (Pan et al., 2024). We extract all the functions (including standalone functions and class methods) from the 358 repositories and sample 2,000 of them. For each target, we use o3-mini (OpenAI, 2025b) to generate a brief description while explicitly avoiding mentioning the function name, file path, or other identifiers.

To initialize the environment for the agent, we run each instance in a base `Python-3.11` Docker container. We clone the repository at the specified commit and remove the target function's docstring.

The agent receives only the functionality description within our prompt shown below:

```
Function Localization Prompt

I've uploaded a python code repository in {workspace_dir}.
Your goal is to locate the {module_type} described below and write its docstring.

<function_description>
{brief_description}
</function_description>

Known details:
- Target type:  {function|class}
- The original docstring was removed; please restore/author it.
- Note:  The target name is not provided.  You must identify the correct
{module_type} based on the functionality description above.

Follow these steps:
1.  EXPLORATION: Search the codebase to find matching functions.
2.  UNDERSTANDING: Read the implementation to understand arguments, return values,
and side effects.
3.  RECHECK: Verify the located function matches the description.
4.  GENERATION: Insert a Python docstring using triple quotes.
5.  REVIEW: Check formatting and accuracy.
```

**Evaluation**. We extract the git diff to parse which and how the files and functions were edited. Success requires: (1) the correct target function is edited, and (2) all changes made to the repository are docstrings/comments only.

**Training Instance Rollout**. We use both Claude-sonnet-3.7 and Qwen3-235B-A22B-Instruct-2507 to generate trajectories and only keep the successful ones in our training set. We started from 2,300 SWE-Gym-Raw functions and finally obtained 1,438 function-localization instances.

### A.1.2. ISSUE LOCALIZATION

**Task**. Given a GitHub issue description, the agent must (1) identify files, classes, functions, and lines of code which are relevant to resolving the issue, and (2) generate comments at the corresponding locations that contain the plan to resolve the issue. The agent should only generate comments and should not modify any actual code.

**Task Implementation**. To implement the task, we use the SWE-Gym-Raw dataset (Pan et al., 2024) as the input of our data, which contains 64,689 GitHub issues sampled from 358 open-source Python projects and the gold patches to resolve them. We use the files that the gold patches locate in as the ground truth locations.

When applying the agents to resolve the tsak, we run each instance in a fresh 'python-3.11' Docker container, with the repository cloned at the corresponding commit. Note that we do not install external packages for the repositories, so only one docker is required for generating all the issue-localization instances.

Below is the prompt provided to the agent:

---

**Issue Localization Prompt**

```
Consider the following issue description:
<issue_description>
{problem_statement}
</issue_description>

Your objective is to localize the specific files, classes or functions, and lines
of code that need modification or contain key information to resolve the issue.

Follow these steps:
1.  Categorize and Extract Key Problem Information
2.  Locate Referenced Modules (use format:  file_path:QualifiedName)
3.  Analyze and Reproduce the Problem
4.  Locate Areas for Modification

Output Format:  List locations in triple backticks with file path, class, function,
and line numbers.  Include about 5 files.

IMPORTANT: You should NOT modify any files!
```

---

**Evaluation**. We extract localization predictions from the code patches generated by the agent and check whether they cover the file where the ground truth patch locates in.

**Training Instance Rollout**. We use the combination of Claude-Sonnet-4.5 and Claude-Sonnet-3.7 to generate trajectories and only keep the successful ones in our training set. We started from 3,000 SWE-Gym-Raw issues and finally obtained 1,978 issue-localization instances.

### A.1.3. DEPENDENCY SEARCH

**Task**. Given a function in a codebase, the agent must identify all functions or classes that it directly calls and add comments to the corresponding module definitions. The agent should only add comments and should not modify any actual code.

**Task Implementation**. We build the tasks based on the SWE-Gym-Raw dataset (Pan et al., 2024). For each unique repository and commit, we parse all Python files and locate candidate target functions. We use an AST-based pass to extract call sites from the function body, and use `Jedi`, a static Python analysis tool, to resolve each call to its actual definition across files by following import chains. We filter out Python built-ins and external packages, and only keep dependencies defined within the repository. We then sample target functions with 1–3 direct dependencies (configurable) and use a balanced sampling scheme across dependency counts, with a cap per repository, to create the dataset.

To initialize the environment for the agent, we run each instance in a base `Python-3.11` Docker container. We clone the repository at the specified commit.

Below is the prompt provided to the agent:

---

**Dependency Search Prompt**

```
I've uploaded a python code repository in {workspace_dir}.
Your task is to analyze the function '{func_name}' located at line {line_number} in
'{file_path}' and find ALL modules (functions or classes) that are directly called by
this function within this repository.

For each called module that is defined within this repository (not Python built-ins
or external libraries), you need to add a comment right above its definition:
'# this function/class is called by the {func_name} function'

What counts as a "call":
- Function calls:  'helper_function()'
- Class instantiation:  'MyClass()'
```

---

```
- Exception raising: `raise MyException()`

Follow these rules:
1.  Only annotate modules that are directly called by `{func_name}` (not transitive
dependencies).
2.  Only annotate modules defined within this repository; ignore Python built-ins,
standard library functions, and external package functions.
3.  Place the comment on the line immediately above the `def` or `class` keyword.
4.  If there are decorators, place the comment above the decorators.
5.  Do NOT modify any code other than adding these comments.
6.  The comment must say exactly:  `# this function/class is called by the
{func_name} function`.
7.  Do NOT add duplicate comments; each dependency should be annotated exactly once.

Follow these steps:
1.  READ: First, read the function `{func_name}` in `{file_path}` to understand what
it does.
2.  ANALYZE: Identify all function/class calls within the function body (typically
1--3 dependencies).
3.  RESOLVE EACH CALL: For each call, determine which definition is actually in
scope by checking local definitions and imports.
4.  FILTER: Exclude Python built-ins and external library calls.
5.  ANNOTATE: Add the comment above each qualifying definition in the repository.
6.  VERIFY: Double-check that all comments are correctly placed and there are no
duplicates.
```

**Evaluation**. We parse the git diff to identify added comments and match them to the expected dependency definitions. A dependency is counted as correct if a comment that mentions the target function is placed above the dependency definition (within a small line-offset tolerance). Success requires: (1) all dependencies are annotated, (2) no extra or duplicate comments are added, and (3) all changes are comments only.

**Training Instance Rollout**. We use Qwen3-235B-A22B-Instruct-2507 to generate the trajectories and only keep the correct ones in our training set. We started from the SWE-Gym-Raw repositories and finally obtained 4,000 dependency-search instances. We sampled 800 instances for rollout and finally obtained 502 successful trajectories for training.

### A.1.4. FUNCTION GENERATION

**Task**. Given a function's signature and docstring (with the body removed), the agent must implement the complete function body.

**Task Implementation**. We build the tasks from the RepoST dataset (Xie et al., 2025b), which contains 7,415 Python functions that have executable test cases across 1,049 repositories. For each instance, the function body is replaced with `pass # TODO: Implement this function`, preserving the signature and docstring.

Each instance runs in a Docker container with the repository at the corresponding commit. The target function would be masked properly before applying the agent to solve the task.

Below is the prompt we use:

---

**Function Generation Prompt**

```
I've uploaded a python code repository in {workspace_dir}.
Your task is to implement the body of the function `{func_name}` in `{file_path}`.

The function signature and docstring are already in the file.  The body has been
replaced with `pass # TODO: Implement this function`.

<function_info>
- File:  {file_path}
- Function:  {func_name}
```

---

```
- Docstring: {docstring}
</function_info>

Follow these steps:
1.  NAVIGATE: Open the file and locate the function.
2.  CONTEXT: Read surrounding code for imports, patterns, and style.
3.  IMPLEMENT: Replace the placeholder with a complete implementation.
4.  VERIFY: Review for correctness and style.
```

**Evaluation**. We extract the implemented function body from the code patch generated by the agent, rename it to `${func_name}_new_implementation`, and execute RepoST's test harness to validate correctness. RepoST provides an evaluation script for each instance, which contains a test function that compares the output of the agent's implementation with the output of the original function. We execute all the evaluation scripts in the docker environment they provide. Note that all the scripts are executed in a single docker.

**Training Instance Rollout**. We use both Claude-Sonnet-3.7 and Qwen3-235B-A22B-Instruct-2507 to generate trajectories and only keep the successful ones in our training set. We started from 2,000 RepoST instances and finally obtained 1,287 function-localization instances.

| Category | Func-Localize | Issue-Localize | Dep-Search | Func-Gen | SWE-Bench | SWT-Bench | Commit-0 |
|---|---|---|---|---|---|---|---|
| Analysis and reasoning | 1.6% | 9.3% | 6.3% | 0.9% | 6.0% | 11.4% | 1.3% |
| Repository exploration | 77.9% | 66.2% | 55.5% | 44.6% | 49.1% | 40.3% | 27.3% |
| Execution of existing code | 0.1% | 0.3% | 0.1% | 0.2% | 4.7% | 6.3% | 3.7% |
| Solution implementation | 17.4% | 23.9% | 35.3% | 45.3% | 12.7% | 16.2% | 53.8% |
| Solution verification | 3.0% | 0.3% | 2.8% | 9.0% | 27.6% | 25.9% | 10.9% |

*Table 6.* We categorize the agent's actions based on the intermediate components of the task, where the categories are pre-defined.

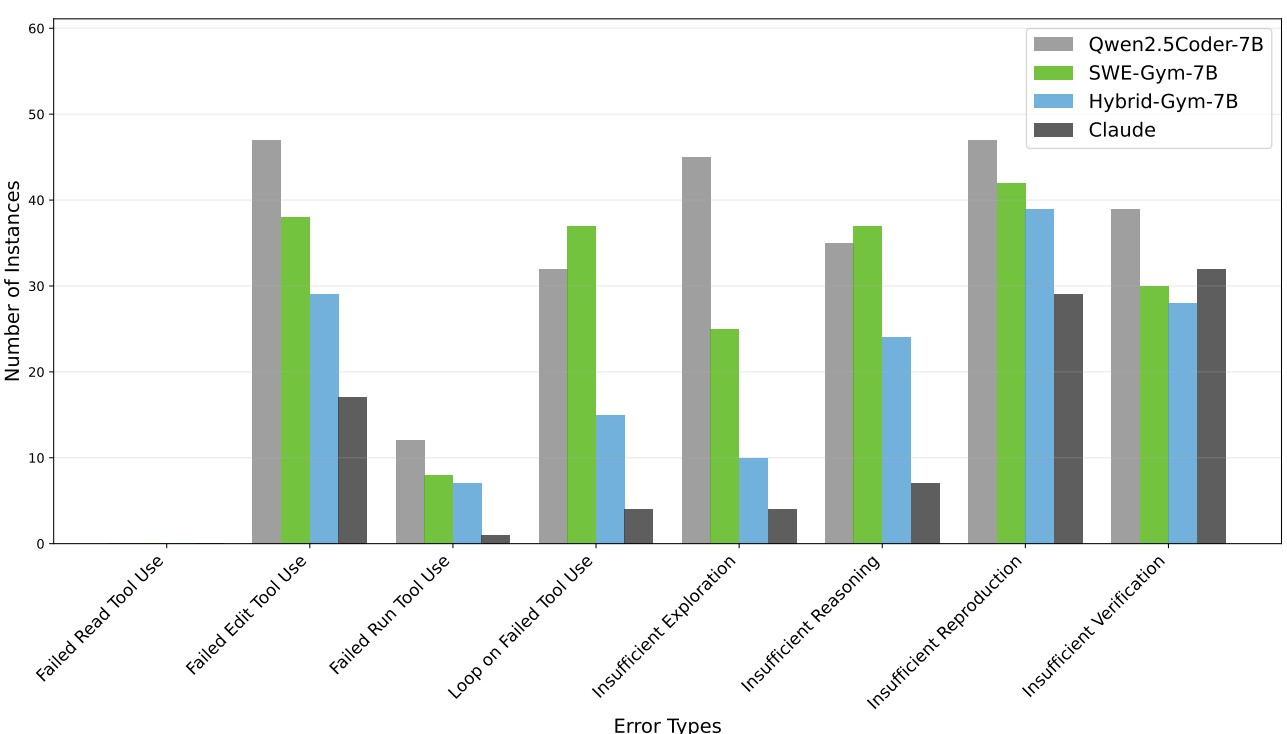

*Figure 5.* The error analysis of different methods on 50 instances sampled from SWE-Bench Verified.

## A.2. Detailed Analysis Results

Table 6 shows the detailed numbers from Figure 1: the percentage of agent steps spent on each component in diverse coding agent tasks.

Figure 5 shows the error analysis results of the base student model (Qwen2.5Coder-7B), the model finetuned on issue-solving (SWE-Gym-7B), the model finetuned on our synthetic tasks (i.e., under the task transfer setting) (HYBRID-GYM-7B), and the teacher model (Claude-Sonnet-3.7). We show the results on SWE-Bench Verified (Easy 50).

## A.3. Experimental Details

### A.3.1. AGENT FRAMEWORK

We use the OpenHands framework (Wang et al., 2025), which provides a sandboxed Docker environment where agents execute bash commands and file operations through a CodeAct interface. For each instance: (1) a Docker container is initialized with the repository at the specified commit, (2) task-specific preprocessing is applied (e.g., removing docstrings or masking function bodies), (3) the agent interacts with the environment, and (4) changes are captured as a git diff.

| Dataset | execute-bash | view | str-replace |
|---|---|---|---|
| Func-Localize | 5.55 | 4.43 | 1.16 |
| Issue-Localize | 9.37 | 7.76 | 5.15 |
| Dep-Search | 1.78 | 3.12 | 1.89 |
| Func-Gen | 0.53 | 1.75 | 1.47 |
| SWE-Bench | 10.65 | 6.88 | 3.65 |
| SWT-Bench | 16.54 | 6.09 | 5.16 |
| Commit-0 | 18.40 | 21.20 | 36.40 |

*Table 7.* The average number of OpenHands tools called in each trajectory.

### A.3.2. TRAINING DETAILS

To train 7B models, we use 8 A6000 GPUs and search hyper-paramers in the range of learning-rate={5e-5, 1e-4}, num-epochs={3,5}, and batch-size={8,16}. The Qwen2.5Coder-7B-Instruct + HYBRID-GYM and Qwen2.5Coder-7B-Instruct + HYBRID-GYM + SWE-Play results we report in Table 4 are both trained with learning-rate=5e-5, num-epochs=5, and batch-size=8. To train Qwen2.5Coder-32B-Instruct, we use 2 H100 GPUs. Due to computational constraints, we use learning-rate=5e-5, num-epochs=5, and batch-size=16 for all the models.

