# OpenReview forum: "Hybrid-Gym: Training Coding Agents to Generalize Across Tasks"
_ICML.cc/2026/Conference — ICML 2026 regular_

### Official Review · Reviewer_TKa2 · 2026-03-11

**Soundness:** 3
**Presentation:** 4
**Significance:** 2
**Originality:** 3
**Overall Recommendation:** 4
**Confidence:** 3

**Summary:**

Instead of training coding agents directly on GitHub issue solving, the authors decompose coding trajectories into transferable skills and train models on scalable synthetic tasks (HYBRID-GYM) that improve real-world coding benchmarks.

**Compliance With Llm Reviewing Policy:**

Affirmed.

**Key Questions For Authors:**

While the paper presents an interesting empirical study and a practical training environment for coding agents, the methodological novelty appears limited. The main contribution lies in the design of synthetic tasks rather than a fundamentally new learning algorithm or theoretical insight.  What is the key methodological novelty of this work that makes it suitable for ICML?

**Limitations:**

Yes.

**Strengths And Weaknesses:**

Strengths

1. The authors attempt to ground the design of synthetic tasks in empirical analysis of real agent trajectories.

2. The empirical improvements are promising

Weaknesses

1. The core idea of decomposing coding tasks into transferable skills and training agents on synthetic auxiliary tasks is intuitive but not fundamentally novel. Similar paradigms have been widely explored in auxiliary task learning, multi-task learning, and hierarchical reinforcement learning. The main contribution appears to lie in the specific task design rather than introducing a new learning framework.

2. The observation that coding agents frequently rely on repository exploration commands (e.g., grep, ls, find) appears somewhat expected and may not provide deep insight into the structure of coding tasks.

3. In experiment, it remains unclear whether the gains stem from the proposed task design or simply from additional training data. The paper does not sufficiently control for dataset size or training compute, making it difficult to isolate the contribution of HYBRID-GYM.

---

> ### Author Rebuttal · Authors · 2026-03-31
>
> We are sincerely grateful for recognizing our contributions of the empirical analysis, practical training environments, and strong empirical results.
>
> &nbsp;
> ## Weakness 1 & Question: Main contribution
> We agree that our work is centered on task design, and we would like to clarify our contributions as follows:
> 1. We are the first work that studies the generalization between coding agent tasks, which opens a broad research direction. Future work may conduct deeper analyses on the transferability of coding agent skills, design a wider set of training tasks, improve the training paradigm specifically for task generalization, etc.
> 2. We show that effective training tasks don't have to have execution or resemble issue solving, test generation, etc., which is a surprising and practical finding.
> 3. Training task design is a non-trivial process, and we don’t just present specific tasks, but also present reusable task design principles, which reveal which tool usage abilities, high-level problem-solving strategies, or concrete agent commands can be transferred/need to be transferred. In addition, our skill-grounded analysis methods that give us insights for training task design could potentially be applied to agent training in other domains (e.g., math or web agents).
>
> &nbsp;
> ## Weakness 2: The observation that different tasks share repository exploration commands is not surprising
> We agree that it might be expected that most coding tasks have repo-exploration as an intermediate step. However, it is still unknown: (1) whether repo-exploration is an important stage for coding agents, and (2) to what extent can learning repo-exploration transfer to downstream coding tasks. Both questions are crucial for understanding coding agent training.
>
> To answer the questions, in our work, we empirically show that (1) repo-exploration is still a bottleneck of coding agent performance, and (2) synthetic training tasks largely improve the agent’s repo-exploration ability and lead to significant performance gain.
>
> &nbsp;
> ## Weakness 3: control for dataset size
> Thank you for raising this point. We want to clarify that we provide a scaling-law analysis in Figure 2, which enables comparison to existing datasets with similar scales. For instance, SWE-Gym (491 trajectories, 20.6% resolved rate on SWE-Bench Verified) has a similar size to our 10% subset (520 trajectories, 22.6% on SWE-Bench Verified).
>
> In fact, one major advantage of our tasks is that they are easy-to-scale, with lower cost and much fewer docker images. In other words, **the large scale of our dataset comes from the scalable nature of our training tasks**.

---

> > ### Author Rebuttal · Reviewer_TKa2 · 2026-04-03
> >
> > Thanks for the rebuttal.

---

### Official Review · Reviewer_FDwy · 2026-03-11

**Soundness:** 3
**Presentation:** 3
**Significance:** 3
**Originality:** 3
**Overall Recommendation:** 5
**Confidence:** 3

**Summary:**

This paper presents HYBRID-GYM, a scalable training dataset for coding agents designed to improve cross-task generalization. The authors first decompose real-world coding agent tasks into intermediate components (reasoning, repository exploration, implementation, verification) and analyze agent capabilities on each. From this analysis, they derive four design principles for training tasks: matching downstream output format, requiring repository exploration, demanding non-trivial reasoning, and avoiding complex environment setup. Guided by these principles, they construct four synthetic tasks -- function localization, issue localization, dependency search, and function generation -- that do not require executable repository environments. Experiments on three benchmarks (SWE-Bench Verified, SWT-Bench Verified, Commit-0 Lite) show strong task transfer, with gains of 25.4%, 7.9%, and 5.1% respectively over the Qwen2.5Coder-32B base model. Additional controlled ablations validate the importance of each design principle and reveal insights about output format, task complexity, trajectory length, teacher model choice, and sampling strategy.

**Compliance With Llm Reviewing Policy:**

Affirmed.

**Final Justification:**

The response has addressed the questions I raised. I  raised my rating. Thank you.

**Key Questions For Authors:**

1. The analysis in Section 4.4 reveals that o3-mini trajectories degrade student performance because reasoning and actions are separated into different turns. Have you investigated whether this phenomenon generalizes to other reasoning-heavy teacher models (e.g., DeepSeek-R1), and whether simple post-processing (as you did for o3-mini) consistently recovers transfer quality? If the authors can show that trajectory format normalization reliably fixes this across multiple teacher models, I would raise my score.
2. In Table 5, combining all four HYBRID-GYM tasks (2000 total instances) achieves 15.0% on SWE-Bench, while function localization alone (500 instances) achieves 12.8%. What is the performance when scaling function localization to 2000 instances -- does the task diversity of HYBRID-GYM still provide gains over a single well-chosen task at equal data scale? If the authors can demonstrate that multi-task training provides clear gains over single-task training at matched data scale, I would raise my score.
3. How sensitive are the results to the choice of base model family? All experiments use Qwen2.5Coder variants. Would the same relative gains hold for other model families such as CodeLlama or StarCoder?

**Limitations:**

The authors briefly mention the Python-only scope in passing but do not formally discuss it as a limitation. The paper would benefit from an explicit limitations section addressing: (1) restriction to Python repositories, (2) reliance on a single agent framework, (3) the distillation-only training paradigm, and (4) the potential ceiling effect as base models become stronger. The impact statement mentions no ethical concerns, which seems reasonable for this work.

**Strengths And Weaknesses:**

Strengths:
- The principled task decomposition analysis in Section 2.1 (Table 6, Table 1) provides a well-motivated foundation for training task design, identifying that reasoning, repo-exploration, and implementation are the dominant shared components across diverse coding tasks, which is a useful contribution to the community's understanding of what makes agent training transfer.
- The experimental results are comprehensive and compelling: Table 4 shows HYBRID-GYM achieves strong task transfer across all three benchmarks without any in-domain training data, and even matches or exceeds in-domain datasets like SWE-Gym in some settings. The cost efficiency is notable -- only 0.14 cents per instance with 2 Docker images versus 128 Docker images for SWE-Smith (Table 3).
- The controlled ablation studies in Section 4 (Figures 3 and 4) are thorough and well-designed, systematically validating each of the four design principles. The finding that output format matters critically (Figure 3a, where removing str_replace collapses SWE-Bench performance) and that script-level tasks do not transfer to repo-level tasks (Figure 3b) provide actionable insights for future work.
- The scaling law analysis (Figure 2) demonstrates consistent improvement as training data increases from 5% to 100%, suggesting the approach has not saturated and that the task design methodology enables continued scaling.

Weaknesses:
- The evaluation is limited to Python-only repositories and a single agent framework (OpenHands CodeAct). It remains unclear whether the four design principles and the resulting tasks generalize to other programming languages or agent scaffolds. The authors should discuss this limitation and ideally provide at least a small-scale experiment with a different language or framework.
- The paper relies exclusively on distillation-based supervised fine-tuning from strong teacher models (Claude-4.5, Claude-3.7, Qwen3-235B). There is no exploration of reinforcement learning or other training paradigms that might better leverage the scalable task design. Given that RL-based approaches like SWE-RL (Wei et al., 2025) are concurrent, a discussion of how HYBRID-GYM tasks could serve as RL environments would strengthen the contribution.
- The four proposed tasks, while well-motivated, are somewhat narrowly scoped around localization and code understanding. The paper acknowledges (end of Section 2.3) that other task designs satisfying the principles are possible but defers exploration to future work. Including even one additional task type (e.g., test generation without executable repos, or refactoring tasks) would strengthen the claim that the principles are broadly applicable.
- The comparison with concurrent and prior work could be more thorough. For example, Table 4 shows R2E-Gym results only on SWE-Bench but not on SWT-Bench or Commit-0. Additionally, the paper does not compare against Zhu et al. (2025) -- the concurrent SWE-Play work -- under identical base model and training conditions for the 32B setting, making it harder to isolate the contribution of task design versus data scale.

---

> ### Author Rebuttal · Authors · 2026-03-31
>
> Thanks for acknowledging our contribution of task decomposition analysis, training task design, empirical results, and analytical experiment design! We answer your questions as follows:
>
> ## [Clarification] Weakness 3 (training tasks) and Weakness 4 (baselines)
> We want to first clarify that **we also have a function generation task**, which requires implementing a function inside a repo, and is beyond localization and code understanding.
> Also, in Table 4, we did compare to and outperform R2E-Gym on both SWT-Bench and Commit-0, and we did use the same base models (Qwen2.5Coder, 7B and 32B) as SWE-Play. Please see below (Weakness 4) for a comparison to SWE-Play that controls the number of training steps.
>
> &nbsp;
> ## Question 1: More Insights on Sec. 4.4
> We actually only see this separation behavior in o3-mini’s trajectories, not even in more recent openAI models such as gpt-5-mini.
>
> We do find other training trajectory behaviors that affect student performance, though. For instance, we find that training trajectories without a systematic repo-exploitation strategy also degrade training performance. We mitigate this by prompting the teacher model (Qwen2.5Coder-32B) with detailed repo-exploration strategies in rollout. This improves the student performance from 4% to 8%.
>
> We will add the new results to Sec. 4.4 to provide deeper insight into how trajectory behaviors beyond correctness affect distillation performance.
>
> &nbsp;
> ## Weakness 1,2 & Limitations & Question 3: Limited Experimental Settings
> Due to time constraints, we will include a limitation section and apply our training environments to the mini-SWE-Agent framework in the revised paper.
> Hybrid-Gym can be naturally used for RL. All 4 tasks provide automatic evaluation that can be used as reward signals. For instance, we can define the reward of the function localization task as: 0 if the code patch is not at the correct function, and otherwise the similarity score to the GT docstring.
>
> &nbsp;
> ## Question 2: Single-Task Training
> Note that function localization has the best SWE-Bench and SWT-Bench performances, but performs the worst on Commit-0.
> Under the similar data scale (~2k data), training Qwen2.5Coder-32B on function localization only gives 28.2% on SWE-Bench-Verified and 11.16% on Commit-0-Lite.
> In comparison, training on ~2k Hybrid-Gym data gives 30.6% on SWE-Bench-Verified and 13.10% on Commit-0-Lite. Therefore, this shows the advantage of including a mix of tasks rather than just function localization.
> Due to time constraints, we will add the full scaling curve (500, 1k, 2k) in the revised paper.
>
> &nbsp;
> ## Weakness 4: Controlled Data Size
> We provide a scaling curve in Figure 2 that supports the comparison at a similar data scale.
>
> Particularly, SWE-Play contains 51.3k steps in the training data and achieves 31.2% on SWE-Bench Verified. Our “25% data” ablation has 48.9k steps and achieves comparable performance (28.4%) on SWE-Bench. We note that SWE Play has in domain (issue solving) training data, while our method does not: we believe it is a strength of our method that we can achieve comparable performance to SWE Play even without in-domain data.
>
> As shown in Table 4, further scaling our Hybrid-Gym data allows performing better than SWE-Play, with 32.4% resolved rate on SWE-Bench. Combining Hybrid-Gym and SWE-Play further improves the results to 33.6%.

---

> > ### Author Rebuttal · Reviewer_FDwy · 2026-04-03
> >
> > The author's response addressed my concerns, so I decided to maintain my positive rating.

---

> > > ### Author Response · Authors · 2026-04-03
> > >
> > > Thanks for acknowledging that your concerns have been fully resolved!
> > >
> > > We believe we have answered all your key questions, too, and would appreciate it if you could consider raising the score if the answers indeed make sense to you.

---

### Official Review · Reviewer_fsGV · 2026-03-15

**Soundness:** 4
**Presentation:** 4
**Significance:** 4
**Originality:** 3
**Overall Recommendation:** 6
**Confidence:** 4

**Summary:**

In this paper, the authors analyze the actions of agents on software engineering tasks and conclude that the most representative subtasks can be synthetized at scale without creating installable repositories. The authors then use this conclusion to guide the creation of 4 task types, which they synthetize at scale, leading to a dataset of 5k trajectories.

The authors demonstrate performance gains on issue resolution, issue reproduction and from-scratch library generation, on Qwen 2.5 Coder 7b and 32b when training on these trajectories. The authors also demonstrate increased performance when mixing the proposed dataset with existing domain-specific datasets targeting the evaluatedn tasks. The authors show that the final performance of the agent scales with the number of training examples, that specific elements of the datastet such as output format are crucial, and highlight the role of teacher model and tasks diversity.

**Compliance With Llm Reviewing Policy:**

Affirmed.

**Final Justification:**

After reading the rebuttal I maintain my score.

**Key Questions For Authors:**

* What performance gains are to be expected from performing RL on these tasks?
* Minor note: you say “Under the setting of distillation learning, a training task is useful only if there is a performance gap between the student and teacher models”; this is not exactly true in the case of rejection sampling: you can spend compute to get good coverage and an appropriate number of trajectories even if the success rate is well below 1.
* I would like to know which version of Claude you are using (Opus, Sonnet?)

**Limitations:**

* A limitation of the paper is that the tasks don't include code generation, which is practically useful for a coding assistant. The authors are upfront about this and remark that code execution is only a small part of the considered benchmarks. It is however an important component of how these agents are used today.

**Strengths And Weaknesses:**

## Strengths

* The depth of the analysis in this paper is exceptional. The main argument is motivated by a-priori analysis, which results in a set of design principles that, when implemented lead to datasets offering very good performance improvements across scales on Qwen 2.5.
* Experimental results are detailed and well-presented and provide strong support for the author's conclusions
* The method is simple, novel, cheap and, according to provided analysis, scales. It will very probably be built on by other researchers working on SWE agents.
* The presentation of the paper is very good, it was easy to follow and every claim very well supported by analysis.
* The design of the ablation and diversity/model experiments is very good, and provides valuable insights on the dataset. I especially like the idea of editing SFT data and examining downstream performance.

## Weaknesses

* No obvious weaknesses found for this paper, perhaps experiments on more recent models could have strengthened the results.

---

> ### Author Rebuttal · Authors · 2026-03-31
>
> We are sincerely grateful for the recognition of the depth of analysis, novelty, practical significance, and presentation of our work!
>
> &nbsp;
> ## Weakness & Question 1: more base models and performance of RL
> We agree that comparing to more recent models would make the results stronger. In particular, we choose the current models to be consistent with existing works (SWE-Gym, R2E-Gym, SWE-Smith, and SWE-Play).
>
> We agree that performing RL on Hybrid-Gym is an important direction for future work. It is difficult to estimate without running the actual experiments, though, and we leave the exploration to future research.
>
> &nbsp;
> ## Question 2: Minor note about distillation
> Thanks for pointing out the more precise claim about distillation! We plan to edit the claim in the revised paper: training tasks are more useful when there is a larger performance gap between the student and teacher models. Does that sound more reasonable to you?
>
> &nbsp;
> ## Question 3: Versions of Claude
> We use Claude-Sonnet-3.7 and Claude-Sonnet-4.5.
>
> &nbsp;
> ## Limitation: code generation in training
> We would like to make a clarification that **we do have a code generation task in Hybrid-Gym**: function generation. The goal is to implement a function, and we evaluate the correctness by copying the agent’s solution to a separate evaluation script, which contains the ground truth function, its dependencies, and a test function.
>
> As for code execution, we agree that it is an important ability. To encourage code execution in training trajectories, one easy way to improve Hybrid-Gym is to wrap the evaluation script in an API call format and provide it to the agent during rollout. In this way, the agent can run the test to verify and improve its own solution. Another way is to simply instruct the teacher model to try to install the repository before implementing the function, and write test scripts after implementation. Although we do not particularly install the repository for the agent, it still has access to repo installation and test generation commands.

---

> > ### Author Rebuttal · Reviewer_fsGV · 2026-04-08
> >
> > I thank the authors for their response.
> > A small mistake: I wrote in my assessment:
> >
> > > A limitation of the paper is that the tasks don't include code generation, which is practically useful for a coding assistant
> >
> > I meant _code execution_, sorry for the confusion.
> > The new formulation on distillation sounds good.
> >
> > After reading the rebuttal i maintain my positive score.

---

### Official Review · Reviewer_mQJL · 2026-03-19

**Soundness:** 4
**Presentation:** 4
**Significance:** 4
**Originality:** 4
**Overall Recommendation:** 5
**Confidence:** 4

**Summary:**

This work proposes Hybrid-Gym, a suite of training tasks that generalize across diverse real-world coding agent settings such as issue resolution, test generation, and library implementation.

To do this, the authors first decompose coding workflows into core components: reasoning, repository exploration, implementation, and verification, label the actions from trajectories from various benchmarks, and observe that most of these actions do not require complex executable code environments.

Based on this analysis, they devise 3 core principles for generating tasks that promote generalization. The tasks should have the same output format, involve a meaningful repository-exploration phase, and involve non-trivial reasoning to complete. They then devised a scalable task generator following these principles to have 4 kinds of tasks:
function localization, issue localization, dependency search, and function generation. None of these requires a complex executable environment. They collect 5.2k trajectories from 762 repositories in these formats.

The experiments show that training on Hybrid-Gym, even alone, results in good performance on all 3 code agent benchmarks, and including benchmark-specific/in-distribution training bumps the score further ahead.

Their ablations show how each principle was important for generalization to work, e.g., not having the same output format significantly drops the performance.

**Compliance With Llm Reviewing Policy:**

Affirmed.

**Key Questions For Authors:**

--

**Limitations:**

yes

**Strengths And Weaknesses:**

# Strengths

* Well-motivated and systematic approach to building coding agent training tasks.
* Demonstrates strong generalization and transfer results on 3 different kinds of benchmarks.
* The approach is much cheaper than existing methods.
* The paper has detailed analysis and ablations establishing the need for various design decisions.
* The paper is clearly written and easy to follow.

# Weaknesses

This is an excellent work. I do not see any major weaknesses that would warrant rejecting the paper.

---

> ### Author Rebuttal · Authors · 2026-03-31
>
> Thanks for your positive and supportive comments! We are sincerely grateful for recognizing the contribution of our method, analysis, and empirical results.

---

### Decision · Program_Chairs · 2026-04-30

**Decision:**

Accept (regular)

**Comment:**

This paper proposes Hybrid-Gym, a synthetic dataset and a set of principled workflows for constructing scalable training tasks to improve the cross-task generalization of coding agents.

All reviewers consistently rated this paper in the accept range, and I strongly concur with their positive assessments. The core strength of this work, as highlighted by the reviewers, lies in its solid theoretical grounding—the decomposition of workflows into shared intermediate components is both intuitive and effective. Furthermore, the empirical results are compelling; the robust ablation studies successfully validate the proposed design principles, and the consistent performance gains across downstream benchmarks demonstrate high practical value.

During the rebuttal, the authors satisfactorily clarified the remaining questions regarding the task design and evaluation setup, effectively resolving the minor concerns raised. Given the strong reviewer consensus, the convincing empirical backing, and the cost-efficient pathway this work provides for the community, I am confident in recommending this paper for acceptance.